# Improvement of Glycaemia and Endothelial Function by a New Low-Dose Curcuminoid in an Animal Model of Type 2 Diabetes

**DOI:** 10.3390/ijms23105652

**Published:** 2022-05-18

**Authors:** Sara Oliveira, Tamaeh Monteiro-Alfredo, Rita Henriques, Carlos Fontes Ribeiro, Raquel Seiça, Teresa Cruz, Célia Cabral, Rosa Fernandes, Fátima Piedade, Maria Paula Robalo, Paulo Matafome, Sónia Silva

**Affiliations:** 1Coimbra Institute of Clinical and Biomedical Research (iCBR), Faculty of Medicine and Center for Innovative Biomedicine and Biotechnology (CIBB), University of Coimbra, 3000-548 Coimbra, Portugal; saraoliveira116@gmail.com (S.O.); tamaehamonteiro@gmail.com (T.M.-A.); celia.cabral@fmed.uc.pt (C.C.); rcfernandes@fmed.uc.pt (R.F.); sonias@ci.uc.pt (S.S.); 2Institute of Physiology, Faculty of Medicine, University of Coimbra, 3000-548 Coimbra, Portugal; rseica@fmed.uc.pt; 3Clinical-Academic Center of Coimbra (CACC), University of Coimbra, 3000-548 Coimbra, Portugal; cribeiro@fmed.uc.pt; 4Research Group on Biotechnology and Bioprospecting Applied to Metabolism (GEBBAM), Federal University of Grande Dourados, Dourados 79825-070, MS, Brazil; 5Faculty of Pharmacy, University of Coimbra, 3000-548 Coimbra, Portugal; ritamahenriques@gmail.com (R.H.); trosete@ff.uc.pt (T.C.); 6Institute of Pharmacology and Experimental Therapeutics, Faculty of Medicine, University of Coimbra, 3000-548 Coimbra, Portugal; 7CNC—Center for Neuroscience and Cell Biology, University of Coimbra, 3004-504 Coimbra, Portugal; 8CQE, Complexo I, Instituto Superior Técnico, University of Lisbon, 1049-001 Lisbon, Portugal; mdpiedade@fc.ul.pt (F.P.); mprobalo@deq.isel.ipl.pt (M.P.R.); 9Faculty of Sciences, University of Lisbon, 1749-016 Lisbon, Portugal; 10Instituto Superior de Engenharia de Lisboa (ISEL), Instituto Politécnico de Lisboa, 1959-007 Lisbon, Portugal; 11Instituto Politécnico de Coimbra, Coimbra Health School (ESTeSC), 3046-854 Coimbra, Portugal

**Keywords:** curcuminoid, endothelial dysfunction, glycaemia, oxidative stress, type 2 diabetes mellitus

## Abstract

Curcumin has been suggested as a promising treatment for metabolic diseases, but the high doses required limit its therapeutic use. In this study, a new curcuminoid is synthesised to increase curcumin anti-inflammatory and antioxidant potential and to achieve hypoglycaemic and protective vascular effects in type 2 diabetic rats in a lower dose. In vitro, the anti-inflammatory effect was determined through the Griess reaction, and the antioxidant activity through ABTS and TBARS assays. In vivo, Goto-Kakizaki rats were treated for 2 weeks with the equimolar dose of curcumin (40 mg/kg/day) or curcuminoid (52.4 mg/kg/day). Fasting glycaemia, insulin tolerance, plasma insulin, insulin signalling, serum FFA, endothelial function and several markers of oxidative stress were evaluated. Both compounds presented a significant anti-inflammatory effect. Moreover, the curcuminoid had a marked hypoglycaemic effect, accompanied by higher GLUT4 levels in adipose tissue. Both compounds increased NO-dependent vasorelaxation, but only the curcuminoid exacerbated the response to ascorbic acid, consistent with a higher decrease in vascular oxidative and nitrosative stress. SOD1 and GLO1 levels were increased in EAT and heart, respectively. Altogether, these data suggest that the curcuminoid developed here has more pronounced effects than curcumin in low doses, improving the oxidative stress, endothelial function and glycaemic profile in type 2 diabetes.

## 1. Introduction

Type 2 diabetes mellitus (T2DM) is a chronic heterogeneous metabolic disease and a current major global health concern, characterised by increased blood glucose levels driven by insulin resistance in target tissues and pancreatic β-cell dysfunction [1]. Exposure to chronic hyperglycaemia increases reactive oxygen species (ROS) through five major pathways, namely, the formation of advanced glycation end-products (AGEs) and the expression of its receptors, polyol pathway flux, activation of protein kinase C isoforms, hexosamine pathway and decreased antioxidant defences [2]. Therefore, the alteration of the redox state stimulates the expression of pro-inflammatory cytokines, adhesion molecules and growth factors, which damage capillary function in several organs, especially the kidneys, heart, major arteries and eyes. Such mechanisms are the foundation of the development of microvascular (nephropathy, retinopathy and neuropathy) and macrovascular diabetes complications (acute coronary syndromes and peripheral vascular disease) [3].

Growing evidence suggest natural products as potential therapeutic approaches, taking advantage of their antioxidant and anti-inflammatory effects and low adverse effects. In addition to being recognised worldwide for multiple potential therapeutic applications, curcumin was considered as “Generally Regarded As Safe” by the Food and Drug Administration. A dose of 0.1–3 mg/kg of body weight was classified as acceptable by the Joint Food and Agriculture Organization of the United Nations/World Health Organization Expert Committee on Food Additives [4]. Curcumin is the main polyphenol found in turmeric rhizome (*Curcuma longa* L.) and has considerable biological and pharmacological activities, such as hypoglycaemic, antioxidant, anti-inflammatory, antibacterial, antimutagenic, antiviral and anticoagulant [5]. Several studies have shown the beneficial effects of curcumin in improving T2DM outcomes, ameliorating glucose and lipid metabolism, as well as macro and microvascular complications in clinical [6,7,8,9] and preclinical settings [10,11,12,13]. Such beneficial effects were not only observed in type 2 diabetes, but also Gestational Diabetes Mellitus in humans and animal models [14,15,16,17,18,19,20,21].

Although it has strong intrinsic activity, curcumin is not yet approved as a therapeutic agent due to its low absorption, limited distribution, rapid hepatic metabolism, rapid systemic elimination and clearance, and low chemical stability, which drastically reduce its bioavailability [22]. Consequently, effective therapeutic responses are only achieved with high doses, typically in the range of 80–250 mg/kg in rodent models during 2 to 10 weeks. Such required high doses compromise patients’ compliance and increase the probability of adverse effects. In order to overcome such limitations, medicinal chemistry strategies, such as modifications in the chemical structure, drug-delivery systems (nanoparticles, phospholipid complexes, liposomes and micelles) and the use of co-adjuvants, have been used to improve the solubility and prevent hepatic transformation [5]. However, despite all efforts to enhance curcumin pharmacological and biological potential, there are still problems in achieving the balance between potency, efficacy and/or bioavailability, while increasing its efficacy in the treatment of diabetes and its complications [22].

In this work, we synthesise a new curcuminoid, which is evaluated for its hypoglycaemic and vascular protective effects in a low-dose treatment of Goto-Kakizaki (GK) rats, an animal model of type 2 diabetes.

## 2. Results

### 2.1. In Vitro Antioxidant and Anti-Inflammatory Effects of the Curcuminoid

In the Alamar blue assay, no significant differences were observed between groups (Figure 1B), suggesting that the curcuminoid has no toxic effects on fibroblasts. Regarding the anti-inflammatory activity, LPS-induced NO production by the Raw 264.7 macrophages was significantly reduced by both the curcumin and the curcuminoid (*p* < 0.001 vs. LPS and LPS + Vehicle; Figure 1C), indicating that both have a marked anti-inflammatory activity. In the TBARS assay, none of the molecules (curcumin and curcuminoid, 25 and 50 µM) were able to reproduce the antioxidant activity of the positive control BHT, both in the absence and presence of the radical inducer ABAP (Figure 1D). In the ABTS assay, curcumin (1 mg/mL) mimicked the antioxidant activity of the positive control’s ascorbic acid and Trolox, while the curcuminoid had less evident effects (Figure 1E).

### 2.2. The Curcuminoid Has Marked Hypoglycaemic Effects

Body weight did not significantly change among the different groups studied, although caloric intake was decreased in Vehicle (*p* < 0.05) and Curcumin (*p* < 0.01) groups compared to the GK group (Figure 2A,B). The area under the curve of ipITT was significantly higher in GK (*p* < 0.001) and Vehicle (*p* < 0.01). The Curcumin- and the Curcuminoid-treated rats had a reduction in AUC, shown by the reduction in the difference in relation to the Wistar Control group (Figure 2C). Moreover, the curcuminoid produced a sustained decrease in the fasting plasma glucose levels (percentage of the initial values) along the experimental period, which was not observed for curcumin (*p* < 0.01 vs. Vehicle; *p* < 0.05 vs. Wistar; Figure 2D). Such an improvement was coincident with increased GLUT4 levels in epididymal adipose tissue (*p* < 0.05 vs. GK and Vehicle; Figure 2H) and plasma insulin levels, which were higher than in the Vehicle group (*p* < 0.05; Figure 2E). Curcumin treatment also increased adipose tissue GLUT4 levels (*p* < 0.05 vs. GK and Vehicle; Figure 2H), while no changes were observed in the insulin-receptor levels and phosphorylation in any of the studied group.

A decrease in the free fatty acid (FFA) levels was observed after the treatment with the curcuminoid (*p* < 0.05 vs. Vehicle; *p* < 0.01 vs. Curcumin; Figure 2F), which was coincident with a partial improvement of epididymal adipose tissue PPARγ levels, reversing the trend to lower levels in GK (*p* < 0.06 vs. Wistar) and Vehicle (*p* < 0.1 vs. Wistar) groups (Figure 2G). No differences were observed in the AMPK levels and its phosphorylation. Such results suggest that, even at such low dose, the curcuminoid possesses hypoglycaemic and hypolipidemic activities that surpass those observed for curcumin in the same dose and may rely on the modulation of glucose uptake and lipid metabolism.

### 2.3. The Curcuminoid Improves Endothelial Function and Oxidative Stress in Type 2 Diabetes

Compared to Wistar control rats, the 12-week-old GK rats showed 7% lower endothelium-mediated relaxation of noradrenaline-precontracted aorta rings in response to ACh, although statistical significance was not observed (Figure 3A and Table 1), and also the vehicle did not produce any significant changes (Figure 3B and Table 1). Vasorelaxation of noradrenaline-precontracted aorta rings in response to ACh was significantly improved by 18 and 12% in Curcumin and Curcuminoid groups, compared to control GK rats (*p* < 0.01 and *p* < 0.05, respectively; Figure 3C,D; Table 1). Detailed data on the maximum effect (E_max_) values are summarised in Table 1. Preincubation of the arterial rings with the eNOS inhibitor L-NG-Nitroarginine Methyl Ester (L-NAME) almost completely abolished relaxation by ACh, showing the endothelium dependence (data not shown).

Preincubation with the antioxidant ascorbic acid increased vascular relaxation in response to growing concentrations of ACh in Wistar rats (Figure 3E). No significant alterations were found in the aortic rings from GK, Vehicle and Curcumin groups incubated in the same conditions (Figure 3F–H). On the other hand, the curcuminoid vasorelaxant response was markedly exacerbated in the presence of ascorbic acid, revealing more sensitivity to antioxidants compared to GK rats, and restoring the response observed in the Wistar control group (*p* < 0.001; Figure 3I).

To investigate the underlying mechanism for the increased aorta relaxation in response to ACh and ascorbic acid in the rats treated with the curcuminoid, the levels of total and phosphorylated eNOS were measured in the aorta. Neither eNOS expression nor its activation were significantly increased in the animals treated with curcumin or the curcuminoid (Figure 4E), suggesting no effects on NO production. Given the increased relaxation after exposure to ascorbic acid in rats treated with the curcuminoid, oxidative stress was evaluated by determining the generation of superoxide through the DHE dye. DHE was increased in diabetic GK rats when compared to Wistar controls (*p* < 0.05; Figure 4C). Although no differences were observed after curcumin treatment, the curcuminoid induced a twofold significant decrease in superoxide staining (*p* < 0.001 vs. GK; *p* < 0.05 vs. Vehicle; *p* < 0.01 vs. Curcumin; Figure 4C). This decline was observed in all layers of the vessel wall, as shown in the representative images (Figure 4A). Accordingly, we investigated whether the decreased superoxide production in aortic-vessel rats treated with the curcuminoid was associated with peroxynitrite formation and the nitration of tyrosine residues. Although no differences were observed in nitrotyrosine staining between Wistar and GK rats, it was significantly inferior in the aorta of diabetic rats treated with the curcuminoid (*p* < 0.05 vs. Wistar and Vehicle; Figure 4D). As previously shown, this decline was also observed in all layers of the vessel wall, as shown in representative figures (Figure 4B). These results indicate that the treatment with the curcuminoid induced a decrease in oxidative stress in intima, media and adventitia, which increased NO bioavailability and vessel relaxation.

### 2.4. The Curcuminoid Modulates Antioxidant Systems in Epididymal Adipose Tissue and the Heart

Given the effects of the curcuminoid in reducing vascular oxidative stress, the pathways involved in the protection against oxidative stress and glycation were investigated in adipose tissue, liver (insulin sensitive), kidney and heart (tissues affected by the diabetic angiopathy). In EAT, the levels of the antioxidant enzymes SOD1 and GLO1 were decreased in the diabetic GK and Vehicle groups compared with the control Wistar group (*p* < 0.05; Figure 5A,B). However, such differences were not observed in the liver (Figure 5C,D). The treatment with the curcuminoid, but not with curcumin, reversed the decline in SOD1 levels observed in the diabetic rats (*p* < 0.01 vs. GK and Vehicle; *p* < 0.05 vs. Curcumin, Figure 5A) and partially improved GLO-1 levels (Figure 5B). No differences were observed in SOD-2 nor catalase levels in both tissues. Moreover, no significant differences were observed in the levels of AGE MG-H1 (Figure 5, central panel), in the histological analysis of both tissues (Figure 5E,F) and in the fluorescence staining of superoxide anion in the liver (Figure 5G). No histological markers of hepatic toxicity were obtained in treated rats (Figure 5F).

In the kidney and heart, two major organs affected by diabetic angiopathy, no differences were observed in SOD-1, SOD-2 and catalase levels in both organs (Figure 6A and representative membranes). However, treatment with the curcuminoid significantly increased GLO1 levels in the heart when compared to Wistar, GK and Vehicle groups (*p* < 0.05), which was not so evident in the curcumin-treated group (Figure 6B). No differences were found in the kidney glomeruli regarding superoxide staining and histological analysis (Figure 6C,D, respectively). No histological markers of kidney toxicity were found in treated rats (Figure 6C).

## 3. Discussion

In this study, we described the hypoglycaemic and protective vascular effects of a new curcuminoid able to mimic and even exceed the beneficial effects of the native molecule in improving diabetes complications in a low-dose regimen. Here, we show that the curcuminoid has marked hypoglycaemic and hypolipidemic effects, also reducing vascular oxidative stress in type 2 diabetic rats. Such effects were associated with a better vascular function, namely, by improving acetylcholine-induced relaxation in the presence of the antioxidant ascorbic acid. Additionally, the curcuminoid contributed to the upregulation of antioxidant defences in organs affected by DM complications. The curcuminoid shows more efficacy than curcumin itself, in a low dose, where curcumin has limited therapeutic potential, and avoids the adverse effects associated with curcumin high dosages.

In this curcuminoid, the positions in the aromatic ring of the hydroxyl and methoxy groups were switched, the hydroxyl being at a meta position and the methoxy group in the para position relative to the central β-diketone chain of the molecule. Moreover, an alkyl chain, in the ester form, was inserted in the α position of the β-keto-enolic moiety. This substitution of the α position has been strongly related to the changes of the tautomeric equilibrium, better kinetic stability in physiological conditions and antitumoral activity with respect to the parent curcuminoids [23]. 

The position change of the -OH group deactivates the hydrogen donation, which, from a biological perspective, provides the molecule a lower antioxidant capacity, since the neutralisation of the peroxyl radical, for example, is directly mediated by the phenolic group. On the other hand, there are ROS, which include peroxide and singlet oxygen, whose neutralisation occurs by a reaction with the diketone chain through the unsaturated bonds of the keto-enol portion, forming pro-oxidising radicals, which are converted and stabilised to less reactive radicals by the phenolic group. In this curcuminoid derivative, the alkyl tert-butyl ester group introduced in the α position of the diketone chain changes the keto/enol ratio, which emphasises that this curcuminoid is more expected to undergo tautomeric interconversion. The NMR spectrum of the curcuminoid shows the presence of both diketo (54%) and keto-enolic (46%) forms in the deuterated methanol solution, while, for curcumin, it only shows the keto-enol form.

Our results show that the direct antioxidant activity was lower than other known antioxidants, namely, BHT, ascorbic acid or Trolox. Indeed, it was even lower than native curcumin, suggesting that other mechanisms of action may be present in vivo than just the intrinsic antioxidant capacity of the molecule. On the other hand, in cell lines, both curcumin and the curcuminoid have shown a strong anti-inflammatory potential.

The beneficial antidiabetic effects of curcumin have been demonstrated in several preclinical trials, using doses in the range of 80–250 mg/kg during 2 to 10 weeks of treatments [5,10,11,12,13,24,25]. Curcumin was able to improve the metabolic profile in male Wistar rats with induced DM and treated for 10 weeks with a dose of 200 mg/kg/day [25], and also with a dose of 90 mg/kg/day for 15 days [24]. Similarly, in high-fat-diet (HFD)-fed C57BL/6J, Pan and collaborators observed an improvement in parameters related to metabolic syndrome after treatment for 8 weeks with 50 mg/kg/day curcumin [12]. Ding et al. investigated similar parameters in HFD-fed C57BL/6J, demonstrating an overall improvement of the metabolic profile, promoted by 40/80 mg/kg/day of curcumin administration for 12 weeks [10]. In addition, antidiabetic effects were also observed in a study using Streptozotocin (STZ)-induced diabetic Sprague Dawley rats fed an HF and glucose diet, in which a dose of 250 mg/kg/day for 8 weeks was used [13]. Furthermore, Zaheri and their colleges treated HFD-fed Wistar rats with 100 mg/day of curcumin for 4 weeks [26]. Curcumin improved the detrimental effect of a high-fat diet via the regulation of STAT3 expression in the skeletal muscle, a key pathway in the metabolism regulation and immune response, which was also observed by Perugini et al. in adipocytes in vitro [26,27,28]. Taking this into consideration, we established a dose of 40 mg/kg/day of curcumin and the equimolar quantity for the curcuminoid (52.4 mg/kg/day), in order to assess the protective effects of this new curcuminoid derivative in a low-dose regimen.

The metabolic profile and vascular complications were assessed using an animal model of T2DM, the GK rat, treated for 14 days with the established doses. In this study, a significantly lower blood glucose with curcumin treatment was not achieved at 40 mg/kg/day, although an equimolar dose of the curcuminoid produced a marked decrease in glycaemia at 7 and 14 days of treatment. As such, the hypoglycaemic effect of the curcuminoid per se was remarkably superior to that of curcumin. Curcumin was also unable to reduce plasma levels of FFA, which may be associated with the use of a low dose when compared with previous studies [12,13,29]. However, a significant decrease in FFA levels was observed with the curcuminoid showing enhanced effects. The excessive accumulation of free fatty acids has repercussions on the insulin signalling pathway, a process named lipotoxicity, that leads to impaired adipokine and cytokine secretion, resulting in a pro-inflammatory environment [30]. The findings of the present study also revealed a significant improvement of GLUT4 in adipose tissue after curcumin and curcuminoid treatments, a protein involved in the glucose uptake. Although not significant, an increase in PPARγ levels was also observed, which is not only important for FFA storage, but also to downregulate pro-inflammatory signals through Nuclear Factor-κB (NF-κB) inhibition. Altogether, also considering the anti-inflammatory properties of the curcuminoid, its enhanced metabolic effects apparently involve a decrease in plasma levels of free fatty acids and an increase in the glucose transporter GLUT4, contributing to an improvement in glucose uptake into the cells. 

In addition to the hypoglycaemic effects, we demonstrated that the curcuminoid plays a crucial role in preventing endothelial dysfunction, which can attenuate the long-term development of vascular complications in diabetes. As previously mentioned, chronic hyperglycaemia-induced oxidative stress, accompanied with other risk factors of diabetes, induces endothelial dysfunction and deteriorates micro- and macrovascular function. The increase in oxidative stress, with a consequent decrease in NO bioavailability, is responsible for altering endothelium-dependent vasodilation, promoting the development of atherosclerosis and vascular complications of T2DM. Diabetic GK rats showed an impaired endothelium-dependent vasorelaxation compared to Wistar rats, in accordance with an enhanced oxidative stress. Endothelial cells are not able to regulate glucose uptake, and long-term hyperglycaemia results in increased ROS production, modulating intracellular pathways that change the production of endothelial-derived relaxing and contracting factors [31]. To understand the role of oxidative stress in the curcuminoid-induced increased vasorelaxation, the vasorelaxation study was repeated with pre-incubation with the antioxidant ascorbic acid. Treatment of GK rats with the curcuminoid significantly decreased vascular oxidative stress and increased acetylcholine-induced NO-mediated vasorelaxation, improving the endothelial function observed in diabetic conditions. Moreover, the curcuminoid effect was markedly superior to that of curcumin in response to antioxidants and in attenuating diabetes-induced endothelial dysfunction and oxidative stress.

Evidence indicates that endothelial dysfunction is strongly related to vascular permeability of the endothelium barrier, which can cause vascular leakage and altered vascular-wall architecture [32,33]. Additionally, this endothelial injury contributes to the development of diabetic microvascular complications [34]. Although no differences were detected in the antioxidant response pathways in the kidney and liver, the antioxidant activity of the curcuminoid was observed in EAT, a tissue that is highly dependent on its vascular function and in which microvascular dysfunction occurs, along with the dysfunction of the tissue itself [35]. There was a normalisation of GLO1 levels and an increase in SOD1 levels in EAT of diabetic rats treated with the curcuminoid. Such effects were also accompanied by an increase in GLO1 levels in the heart, contributing to improve the oxidative state. The modest effects of curcumin and the curcuminoid in these organs may be explained by the limited alterations present in the diabetic rats at this age. While their vascular function is apparently impaired from an early age, other tissues are apparently more conserved over time. These results are in accordance with the previous observations from our group [36,37]. 

Currently available data suggest that curcumin has relevant antidiabetic effects by protecting from DM macro- and microvascular complications, although it has a low bioavailability and high doses are necessary to achieve a therapeutic goal. Our results demonstrate that this curcuminoid used in a low-dose regimen improves the glycaemic and lipid profiles, with a consequent decrease in oxidative stress, which therefore improves vascular function (Figure 7). Thus, this study supports the possibility of developing and applying new chemical strategies to change curcumin and reduce the doses used, while increasing efficacy in the treatment of diabetes and its complications. Future research should also be directed to the development of new delivery strategies able to surpass the low solubility of curcuminoids.

## 4. Materials and Methods

### 4.1. Chemicals and Antibodies

All reagents and solvents are commercially available, purchased with high purity from Sigma-Aldrich (St. Louis, MO, USA) and Alfa Aesar (Tewksbury, MA, USA) and used without further purification. Reagents used in the preparation of the physiological Krebs–Henseleit solution were purchased from Panreac (Castellar del Vallès, Barcelona, Spain). Antibodies used were targeted to glucose transporter type 4 (GLUT4; 1:1000), phospho-Insulin Receptor β (phospho-IR; Y1361; 1:1000), glyoxalase 1 (GLO1; 1:1000) (Ab65267, Ab60946, Ab96032, Abcam, Cambridge, UK), IRβ (1:1000), superoxide dismutase 1 (SOD1; 1:1000), superoxide dismutase 2 (SOD2; 1:1000), Catalase (Ab76110, Abcam, Cambridge, UK; 1:1000), peroxisome proliferator-activated receptor gamma (PPARγ; 1:1000), phospho-AMP-activated protein kinase alpha (phospho-AMPKα; 1:1000) (Thr172), AMPKα (1:1000) (#2443, #2535, #2532, Cell Signalling, Danvers, MA, USA), phospho-endothelial nitric oxide synthase (phospho-eNOS; 1:250) (ps1177), eNOS (612392, 610296, BD Transduction Laboratories, Franklin Lakes, NJ, USA; 1:250) and methylglyoxal-derived hydroimidazolone-1 (MG-H1; 1:50) (HM5017, HycultBiotech, Uden, Netherlands). Calnexin was used as loading control (AB0037, Sicgen, Cantanhede, Portugal; 1:1000).

Reactions were monitored by thin-layer chromatography (TLC) on aluminium sheets pre-coated with silica gel Merck 60 F254. Melting points were determined with an Electrothermal 9100 digital melting-point apparatus. Nuclear Magnetic Resonance (NMR) spectra were recorded on a Bruker Avance (400 MHz) spectrometer in CD_3_OD-d_4_, dimethyl sulfoxide-d_6_ and acetone-d_6_ as solvents with a 5 mm probe. Spectra were referenced to the residual ^1^H signal of the deuterated solvent. The following abbreviations were used to report the NMR data: s (singlet), d (doublet), t (triplet) and m (multiplet). Coupling constants (J) are in Hz. The electronic spectra of the compounds were measured in acetone solutions (2.0 × 10^−4^ M concentration) in quartz cells using a Cary 60 UV-Vis from Agilent Technologies over the range 200–900 nm.

### 4.2. Chemical Synthesis of Curcumin (2) and Curcuminoid (3)

tert-Butyl-3-acetyl-4-oxopentanoate (1) was prepared following a procedure previously reported by Ferrari et al. [38]. Curcumin (2) (Figure 1A) was synthesised by the process described below, a modification of the procedure previously reported by Pabon [39]. A suspension of acetylacetone (2.16 g, 10 mmol) and B_2_O_3_ (0.70 g, 10 mmol) in ethyl acetate (20 mL) was stirred for 30 min at 40 °C and then tributylborate (10 mL, 40 mmol, 4 eq) was added. After 30 min, vanillin (3 g, 20 mmol, 2 eq) was added and followed by the gradual addition of n-butylamine (0.4 mL, 4 mmol in 8 mL of ethyl acetate). After stirring overnight at 40 °C, the solution was acidified with 0.5 M HCl (30 mL) and cooled down to room temperature. The yellow−orange solid was suspended in water, filtered and dried under vacuum. The crude orange solid was recrystallised from ethanol:water.

Curcumin: (1E,6E)-1,7-bis(4-hydroxy-3-methoxyphenyl)hepta-1,6-diene-3,5-dione (2); orange powder; 72% yield; m.p.: 178–179 °C; ^1^H NMR (MeOD-d^4^, 400 MHz): KE (100%): 7.58 (d, 2H, H-4, J = 15.9 Hz), 7.22 (s, 2H, H-6), 7.11 (d, 2H, H-9, J = 8.0 Hz), 6.82 (d, 2H, H-10, J = 8.0 Hz), 6.63 (d, 2H, H-3, 15.9 Hz), 6.00 (s, 1H, H-1), 3.92 (s, 6H, Ar-OCH3); ^13^C NMR (acetone-d^6^): 183.6, 149.9, 146.9, 140.5, 127.3, 122.9, 121.4, 115.31, 110.7, 55.4; UV-vis (acetone): (420 nm, 8 980 M^−1^.cm^−1^).

Curcuminoid (3) (Figure 1A) was synthesised by the process described below. A suspension of (1) (2.16 g, 10 mmol) and B_2_O_3_ (0.70 g, 10 mmol) in ethyl acetate (20 mL) was stirred for 30 min at 40 °C and then tributylborate (10 mL, 40 mmol, 4 eq) was added. After 30 min, isovanillin (3 g, 20 mmol, 2 eq) was added and followed by the gradual addition of n-butylamine (0.4 mL, 4 mmol in 8 mL of ethyl acetate). After stirring overnight at 40 °C, the solution was acidified with 0.5 M HCl (30 mL) and cooled down to room temperature. The orange solid was suspended in water, filtered and dried under vacuum. The crude orange–red compound was recrystallised from EtOH:hexane.

(3): tert-butyl(E)-6-(3-hydroxy-4-methoxyphenyl)-3-((E)-3-(3-hydroxy-4-methoxy-phe-nyl)acryloyl)4-oxohex-5-enoate (3); orange-red powder; 75% yield; m.p.: 191–193 °C; ^1^H NMR (MeOD-d^4^, 400 MHz): KE (17%): 7.62 (d, 2H, H-4, J = 16 Hz), 7.14 (s, 2H, H-6), 7.11 (d, 2H, H-9), 6.96 (d, 2H, H-10, J = 8 Hz), 6.84 (d, 2H, H-3, J = 16 Hz), 3.89 (s, 6H, Ar-OCH_3_), 3.58 (s, 2H, H-11), 1.42 (s, 9H, −COO(CH_3_)_3_); DK (83%): 7.65 (d, 2H, H-4, J = 16 Hz), 7.14 (s, 2H, H-6), 7.11 (d, 2H, H-9), 7.08 (d, 2H, H-3, J = 16 Hz), 6.97 (d, 2H, H-10, J = 8 Hz), 4.60 (t, 1H, H-1), 3.91 (s, 6H, Ar-OCH_3_), 2.87 (s, 2H, H-11), 1.44 (s, 9H, −COO(CH_3_)_3_); ^13^C NMR (acetone-d^6^, 400 MHz): d KE: 184.7, 172.0, 148.1, 146.6, 142.2, 128.8, 123.8, 115.4, 115.1, 112.6, 105.1, 82.4, 56.4, 35.2, 28.5; DK: 190.1, 172.0, 148.1, 146.6, 146.2, 128.8, 123.8, 123.5, 115.1, 112.6, 82.4, 62.0, 56.4, 38.6, 28.5, UV-VIS (acetone): (430 nm, 6 912 M^−1^.cm^−1^).

### 4.3. Cell Culture

A fibroblast cell line derived from a green monkey (Cercopithecus aethiops) kidney (Cos-7) was cultured in Dulbecco’s modified Eagle medium-high glucose (DMEM-HG), supplemented with 10% fetal bovine serum (FBS) and 1% penicillin/streptomycin at 37 °C, in a humidified incubator with a 5% carbon dioxide (CO_2_) atmosphere. 

RAW 264.7, a mouse leukemic macrophage cell line (ATCC TIB-71), was cultured in DMEM supplemented with 10% (*v*/*v*) of non-inactivated FBS, 3.02 g/L sodium bicarbonate, 100 U/mL penicillin and 100 μg/mL streptomycin at 37 °C, in a humidified atmosphere with 5% carbon dioxide.

### 4.4. In Vitro Cell Viability in Cos-7

The assessment of metabolically active cells was performed using resazurin colorimetric assay [40]. Briefly, Cos-7 cells were seeded and allowed to stabilise in 96-well plates at a density of 8 × 10^4^ cells/mL. Following 24 h, the cells were either maintained in culture medium (control) or incubated with 5 µM of curcumin and the curcuminoid during 24 h. After this period, a solution of DMEM-HG with resazurin was added to a final concentration of 0.1 mg/mL. Dimethyl sulfoxide (DMSO) was used as vehicle in a concentration of 0.5%. The bioreduction in the dye was quantified after 4 h incubation by absorbance measurements at 570 nm and 600 nm in a microplate reader (BioTek Instruments, Inc., Winooski, VT, USA).

### 4.5. Griess Reagent Method for NO Detectionin RAW 264.7

To assess the production of nitric oxide (NO) through the accumulation of its stable product nitrite in the macrophage cell line culture supernatants, a colorimetric reaction based on Griess reagent was used [41]. Briefly, RAW 274.7 at a density of 3 × 10^5^ cells/mL was pre-treated with 5 µM of curcumin or the curcuminoid and pre-incubated with the Toll-like receptor agonist lipopolysaccharide for 24 h. Following this period, cell culture medium was mixed with Griess reagent (1:1) and incubated for 10 min at room temperature. DMSO was used as a vehicle in a concentration of 0.5%. The amount of nitrite in the media was measured by absorbance measurement at 550 nm in a BioTek microplate reader (BioTek Instruments, Inc., Winooski, VT, USA) and calculated from a sodium nitrite (NaNO_2_) standard curve.

### 4.6. In Vitro Antioxidant Activity

#### 4.6.1. 2,2′-Azino-bis(3-Ethylbenzothiazoline-6-Sulfonic Acid) (ABTS) Enzymatic Assay

A mixture of a solution of ABTS (3 mL, 1.94 mM) and a solution of MnO_2_ (3 mL, 75 mg, 0.29 M, 150 eq) prepared in phosphate buffer (pH 7, 0.1 M) was shaken at room temperature for 10 min. The excess of MnO_2_ was removed by centrifugation and filtration. The resulting blue–green solution of the radical ABTS^+^ was kept at 4 °C until its use and the absorbance (A_control ABTS_) was recorded at λ_max_ 734 nm. The absorbance (A_sample_) was measured upon the addition of 20 µL of 1 mg/mL solution of the curcumin or the curcuminoid in spectroscopic grade EtOH to the ABTS solution, after 6 min of reaction. The decrease in absorbance is expressed as % inhibition, which is calculated from the equation:(1)$$\%\;\mathrm{ABTS}\;\mathrm{inhibition}\;=\frac{A_{\mathrm{Control}\;\mathrm{ABTS}}-A_{\mathrm{sample}}}{A_{\mathrm{Control}\;\mathrm{ABTS}}\;}\;\times 100$$

Trolox and ascorbic acid 20 µL (2 mM) solutions were used as standard antioxidants (positive control). Three independent experiments were performed [42].

#### 4.6.2. Thiobarbituric Acid-Reactive Substances (TBARS) Assay

Malondialdehyde (MDA) levels, a product of lipid peroxidation, were determined with a modified thiobarbituric acid (TBA)-reactive substances (TBARS) assay, without (A) or with (B) a lipid peroxidation inducer [43]. (A) Egg yolk homogenised in 1.15% KCl and sample (curcumin and the curcuminoid, 25 µM and 50 µM) or butylated hydroxytoluene (BHT) as positive control (250 µM) were mixed in a test tube and made up to 1 mL with Milli-Q water. Subsequently, 1.5 mL 20% *w*/*v* acetic acid (pH 3.5) and 1.5 mL 0.8% *w*/*v* TBA in 1.1% *w*/*v* sodium dodecyl sulphate (SDS) was added. This mixture was stirred in a vortex and heated at 95 °C for 60 min. After cooling down and reaching room temperature, 5 mL of n-butanol was added to each tube, vortexed and centrifuged at 4500 rpm for 15 min. TBARS levels were measured at 532 nm using a spectrophotometer (Cintra 101, GBC, Braeside, Australia) in the software Cintral General Applications. (B) To induce lipid peroxidation, 50 µL of 2,2-azobis-(2-amidinopropane) dihydrochloride (ABAP) (0.07 M) were added, and the procedure reported above was repeated.

Data were obtained from three independent experiments in triplicate. The antioxidant index was calculated using the following formula:(2)$$\mathrm{Antioxidant}\;\mathrm{Index}\;\left(\%\right)=\frac{\left(1-S\right)}{C\;}\;\times 100\;\;\;$$
where *S* is the absorbance in the presence of the curcuminoid and *C* is the absorbance in its absence (control).

### 4.7. Animal Maintenance

Male 12-week-old Wistar and Goto-Kakizaki (GK) rats from our breeding colonies (Faculty of Medicine, University of Coimbra) were kept under standard conditions (22.0 ± 0.1 °C, 52.0 ± 2.0% of relative humidity, 12 h light/dark cycle). All procedures involving animals were in accordance with the European Community guidelines for the use of animals in a laboratory (Directive 2010/63/EU) and performed by users licensed by the Federation for European Laboratory Animal Science Association (FELASA).

### 4.8. Experimental Groups

Male GK rats were randomly divided into 4 groups (*n* = 5–9/group): (1) Control; (2) Vehicle (Vh), submitted to DMSO subcutaneous (s.c.) injection for 14 days; (3) Curcumin, submitted to 40 mg/kg/day curcumin administration via s.c. injection for 14 days and (4) Curcuminoid, submitted to 52 mg/kg/day curcuminoid administration via s.c. injection for 14 days. A group of five Wistar rats was used as the non-diabetic control (W). The maximum dose of DMSO usually considered and recommended for the intravenous administration route was 0.1 mL/kg in a percentage of 100% (*v*/*v*) [44]. In this study, the volume of DMSO required for the dilution of the compounds was always below that value and the administration was subcutaneous, a less invasive route with less bioavailability.

### 4.9. In Vivo Procedures and Sample Collection

Body weight and caloric intake were monitored during the treatment. Fasting blood glucose (6 h) were determined in blood from the tail vein. The intraperitoneal insulin tolerance test (ipITT) was performed after a 6 h fast using insulin per body weight (250 mU/kg) and the evaluation of glycaemia was performed at 0, 15, 30 and 60 min using a glucose meter and test strips (Accu-Chek Aviva, Roche, Basel, Switzerland). The area under the curve (AUC) was calculated. After treatment, animals were anesthetised and serum and plasma were collected to Vacuette K3EDTA tubes and Vacuette Z Serum clot activator tubes (Greiner Bio-One, Kremsmünster, Austria) for the measurement of plasma insulin and non-esterified fatty acids using the Rat Insulin ELISA Kit (Mercodia, Uppsala, Sweden) and the Serum/Plasma Fatty Acid Kit (Zenbio, Durham, NC, USA). After blood sample collection, animals were sacrificed by cervical displacement and the following tissues were collected: aorta, epididymal adipose tissue (EAT), liver, heart and kidney. Aorta was excised and used for functional studies, and EAT and liver sections, used for colorimetric staining, were immersed in 10% formalin and included in paraffin. Aorta, kidney and liver sections for fluorescence studies were preserved in OCT cryo-embedding media (Thermo Fisher Scientific, Waltham, MA, USA) and stored at −80 °C.

### 4.10. Functional Studies

Aorta rings were mounted on stainless-steel hooks under 19.6 mN basal tension in organ baths filled with aerated (95% O_2_, 5% CO_2_) Krebs–Henseleit solution (37 °C, pH 7.4) (NaCl 118.67 mmol/L; KCl 5.36 mmol/L; CaCl_2_ 1.90 mmol/L; MgSO_4_ 0.57 mmol/L; NaHCO_3_ 25.00 mmol/L; KH_2_PO_4_.H_2_O 0.90 mmol/L; glucose 11.1 mmol/L). Following a 60 min equilibration period, cumulative isometric concentration-response curves to acetylcholine (ACh) (0.01 to 90 µM) were performed in the precontracted aorta rings with 10 µM of noradrenaline, in the presence and absence of 100 µM ascorbic acid. Cumulative curves were recorded with Letica Scientific Instruments isometric transducers connected to a four-channel polygraph (Polygraph 4006, Letica Scientific Instruments, Barcelona, Spain).

### 4.11. Western Blotting

EAT, heart, liver and kidney (*n* = 5) were homogenised with lysis buffer (0.25 M Tris-HCl, 125 mM NaCl, 1% Triton-X-100, 0.5% SDS, 1 mM EDTA, 1 mM EGTA, 20 mM NaF, 2 mM Na_3_VO_4_, 10 mM β-glycerophosphate, 2.5 mM of sodium pyrophosphate, 10 mM of PMSF and 40 μL of protease inhibitor) and the supernatant was mixed with Laemmli buffer (62.5 mM Tris-HCl, 10% glycerol, 2% SDS, 5% β-mercaptoethanol, 0.01% bromophenol blue). Protein concentration of the lysate was determined using the BCA Protein Assay Kit (BioRad, Hercules, CA, USA). Samples were loaded onto a polyacrylamide gel, separated through electrophoresis and transferred to polyvinylidene fluoride (PVDF) membranes. After blocking (TBS-T 0.01% and 5% BSA), membranes were incubated overnight with primary antibodies followed by 2 h incubation with secondary antibodies (anti-mouse, anti-rabbit and anti-goat) [45]. Immunoblots were revealed with ECL substrate in a Versadoc system (Bio-Rad, Hercules, CA, USA) and analysed with ImageQuant (Molecular Dynamics, Chatsworth, CA, USA).

### 4.12. Histological Colorimetric Assays

Tissue sections (4 µm) of EAT, liver and kidney (*n* = 3) were embedded in paraffin and stained with Periodic Acid-Schiff (PAS), hematoxylin and eosin (H&E) or Masson Trichrome staining [45]. Images were obtained using a fluorescence microscope (Zeiss Axio Observer Z1, Carl Zeiss, Gottingen, Germany).

### 4.13. Assessment of Aortic Immunofluorescence

Aorta sections (6 µm) were fixed in cold acetone for 10 min and then washed with phosphate-buffered saline (PBS), followed by permeabilization for 30 min with 0.25% Triton X-100 in PBS, with 1% BSA for 1 h and with 0.02% BSA in PBS for 30 min. After being blocked, sections were incubated with Anti-Nitrotyrosine Antibody (#06-284, Millipore, Germany) (1:200). Sections were then incubated with secondary fluorescent antibody for 1 h at room temperature and washed [46]. Images were obtained using a fluorescence microscope (Zeiss Axio Observer Z1, Carl Zeiss, Gottingen, Germany), detected with 493 nm excitation and 517 nm emission for nitrotyrosine, and 353 nm excitation and 465 nm emission for DAPI.

### 4.14. Detection of Superoxide Anion

To evaluate the intracellular ROS, aorta, kidney and liver sections were incubated with 2 µM of dihydroethidium (DHE) for 30 min at 37 °C in a humidified atmosphere protected from light, following by washes and incubation with 4′,6-diamidino-2-phenylindole (DAPI; 1:2000) to stain the cell nucleus [46]. Samples were imaged with a fluorescence microscope (Zeiss Axio Observer Z1, Carl Zeiss, Gottingen, Germany) with identical settings. DHE fluorescence was detected with 568 nm excitation and 465 nm emission, and DAPI with 353 nm excitation and 465 nm emission. Fluorescence was quantified using ImageJ software.

### 4.15. Statistical Analysis

The results are expressed as the mean ± standard error of the mean (S.E.M.) per group. Statistical analysis was performed using SPSS software (IBM, Armonk, NY, USA). The normality of the data was assessed with a Shapiro–Wilk normality test. In relation to the parametric data, a parametric One-way Analysis of Variance (ANOVA) was applied, followed by Tukey’s multiple comparisons test. For the non-parametric data, the Kruskal–Wallis test was used; the *p*-value < 0.05 was considered statistically significant.

## Figures and Tables

**Figure 1 ijms-23-05652-f001:**
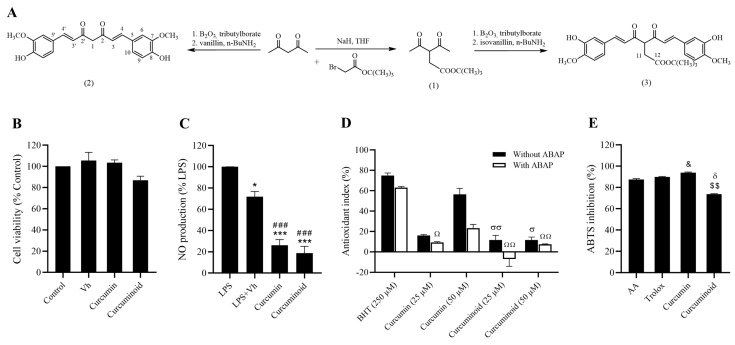
Representative scheme of the chemical synthesis of curcumin (2) and curcuminoid (3) are shown (**A**). Curcumin and curcuminoid (3) demonstrate a strong anti-inflammatory potential in vitro. Cell viability was evaluated in Cos-7 cells by the Alamar blue assay (**B**) and the anti-inflammatory effect was assessed in Raw 264.7 macrophages through the Griess reagent method (**C**). Antioxidant activity was evaluated by TBARS (**D**) and ABTS assays (**E**). Results are expressed as mean ± S.E.M.; vertical bars represent S.E.M.; statistical differences were evaluated by Tukey’s test. * vs. LPS; # vs. LPS + Vh; & vs. AA; δ vs. Trolox; $ vs. Curcumin; σ vs. BHT (250 µM) without ABAP; Ω vs. BHT (250 µM) with ABAP. 1 symbol *p* < 0.05; 2 symbol *p* < 0.01; 3 symbol *p* < 0.001.

**Figure 2 ijms-23-05652-f002:**
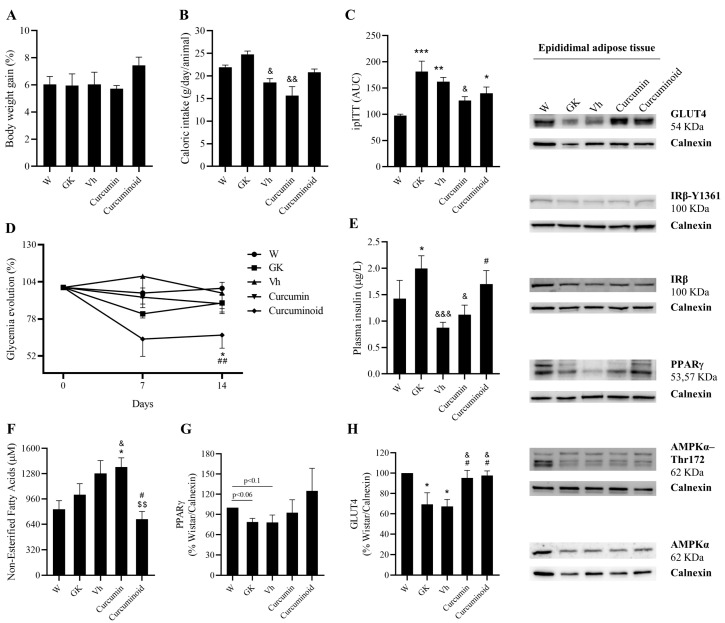
The curcuminoid has marked hypoglycaemic effects, along with an improvement in FFA and GLUT4 levels. Body weight (**A**), caloric intake (**B**), insulin tolerance test (**C**) and glycaemia (**D**) were evaluated during the time course of the treatment. In addition, plasma insulin (**E**) and FFA levels (**F**) were assessed by ELISA kit, and EAT content of PPARγ (**G**) and GLUT4 (**H**) were investigated by WB. W: Wistar control; GK: Goto-Kakizaki control; Vh: Goto-Kakizaki submitted to vehicle (DMSO) administration; Curcumin: Goto-Kakizaki submitted to curcumin administration (40 mg/Kg/day, s.c.) and Curcuminoid: Goto-Kakizaki submitted to curcuminoid (52.4 mg/Kg/day, s.c.) administration. Results are expressed as mean ± S.E.M.; vertical bars represent S.E.M.; *n* = 5–9/group; statistical differences were evaluated by Tukey’s test. * vs. W; & vs. GK; # vs. Vh; $ vs. Curcumin. 1 symbol *p* < 0.05; 2 symbol *p* < 0.01; 3 symbol *p* < 0.001.

**Figure 3 ijms-23-05652-f003:**
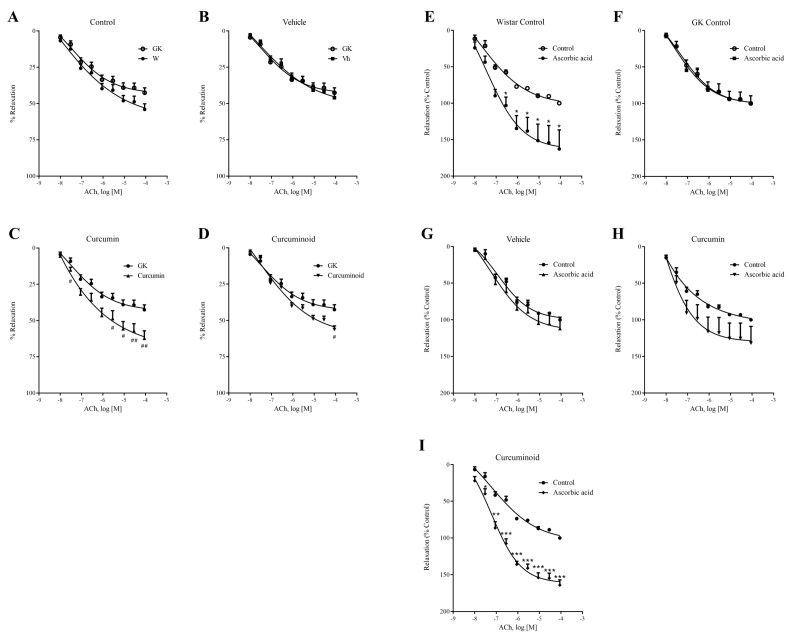
Effects of curcumin or curcuminoid treatment on relaxation response of the isolated aorta to acetylcholine (ACh) before (**A**–**D**) and after preincubation with ascorbic acid (**E**–**I**) from Wistar rats (**A**,**E**), GK control rats (**F**), GK Vehicle rats (**B**,**G**) and GK rats administered with curcumin (**C**,**H**) or curcuminoid (**D**,**I**). W: Wistar control; GK: Goto-Kakizaki control; Vh: Goto-Kakizaki submitted to vehicle (DMSO) administration; Curcumin: Goto-Kakizaki submitted to curcumin administration (40 mg/Kg/day, s.c.) and Curcuminoid: Goto-Kakizaki submitted to curcuminoid (52.4 mg/Kg/day, s.c.) administration. Relaxation is presented as percentage of NA induced contraction; results are expressed as mean ± S.E.M.; *n* = 5–6/group; statistical differences were evaluated by Student *t*-test. # vs. GK; * vs. Control. 1 symbol *p* < 0.05; 2 symbol *p* < 0.01; 3 symbol *p* < 0.001.

**Figure 4 ijms-23-05652-f004:**
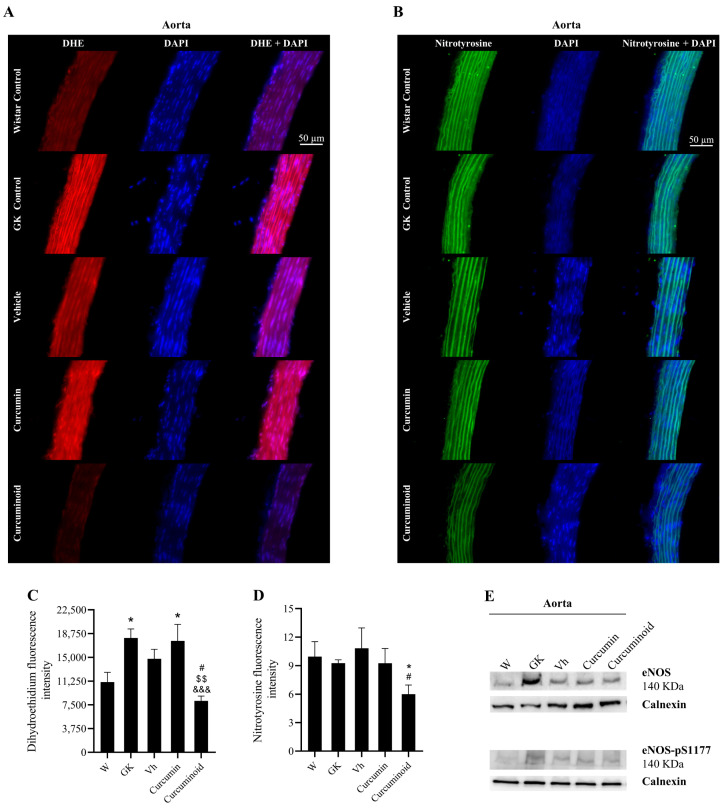
The curcuminoid improved the oxidative and nitrosative stress. Oxidative stress was evaluated though the determination of superoxide generation by the DHE dye. (**A**) Shows representative DHE staining (red) in aortic sections and (**C**) shows quantification of the DAPI (blue)-normalised red fluorescence. Nitrosative stress was evaluated using an anti-nitrotyrosine antibody. Representative aortic sections of nitrotyrosine staining (green) (**B**) quantification of the green fluorescence (DAPI normalised) (**D**). Total and phosphorylated eNOS were assessed by WB; representative WB are shown (**E**); W: Wistar control; GK: Goto-Kakizaki control; Vh: Goto-Kakizaki submitted to vehicle (DMSO) administration; Curcumin: Goto-Kakizaki submitted to curcumin administration (40 mg/Kg/day, s.c.) and Curcuminoid: Goto-Kakizaki submitted to curcuminoid (52.4 mg/Kg/day, s.c.) administration. Results are expressed as mean ± S.E.M.; vertical bars represent S.E.M.; *n* = 5–9/group; statistical differences were evaluated by Tukey’s test. * vs. W; & vs. GK; # vs. Vh; $ vs. Curcumin. 1 symbol *p* < 0.05; 2 symbol *p* < 0.01; 3 symbol *p* < 0.001.

**Figure 5 ijms-23-05652-f005:**
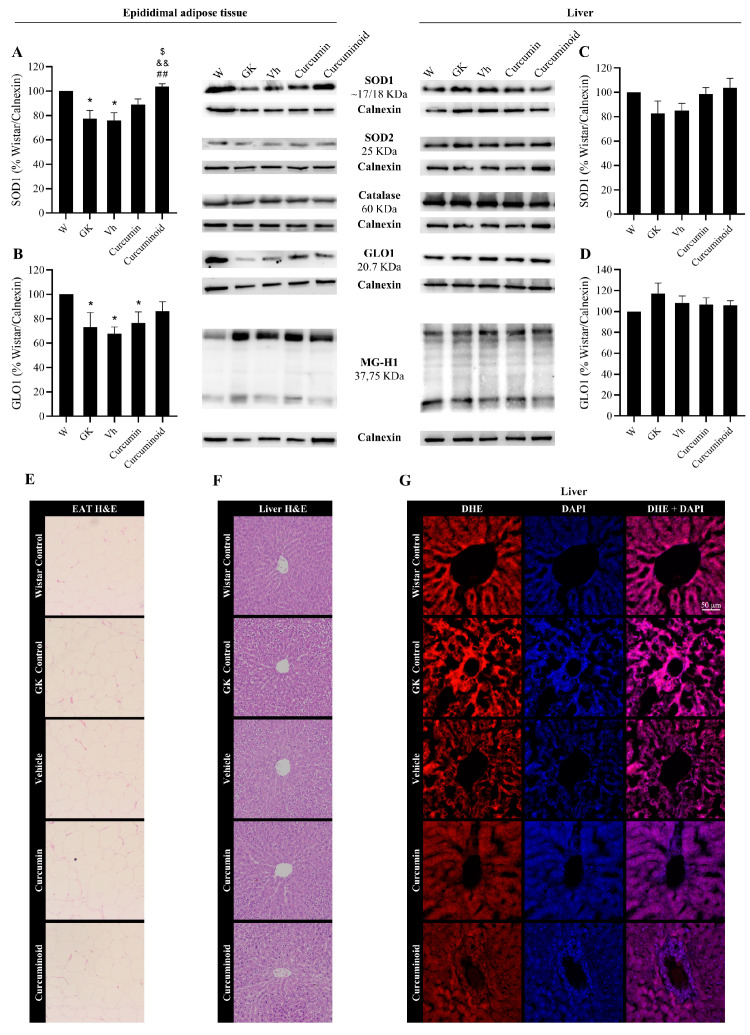
Improved levels of SOD1 (**A**) and GLO1 (**B**) in epididymal adipose tissue, calculated as percentage of Wistar control/Calnexin, evaluated by WB after the treatment with the curcuminoid; representative WB is shown. SOD1 (**C**) and GLO1 (**D**) levels were also evaluated in liver by WB; representative WB is shown. Histological analysis shows hematoxylin–eosin (H&E) staining in EAT (**E**) and liver (**F**). Representative liver sections show the staining (red) of the superoxide-sensitive DHE dye (**G**). W:Wistar control; GK: Goto-Kakizaki control; Vh: Goto-Kakizaki submitted to vehicle (DMSO) administration; Curcumin: Goto-Kakizaki submitted to curcumin administration (40 mg/Kg/day, s.c.) and Curcuminoid: Goto-Kakizaki submitted to curcuminoid (52.4 mg/Kg/day, s.c.) administration. Results are expressed as mean ± S.E.M.; vertical bars represent S.E.M.; *n* = 5–9/group; statistical differences were evaluated by Tukey’s test. * vs. W; & vs. GK; # vs. Vh; $ vs. Curcumin. 1 symbol *p* < 0.05; 2 symbol *p* < 0.01.

**Figure 6 ijms-23-05652-f006:**
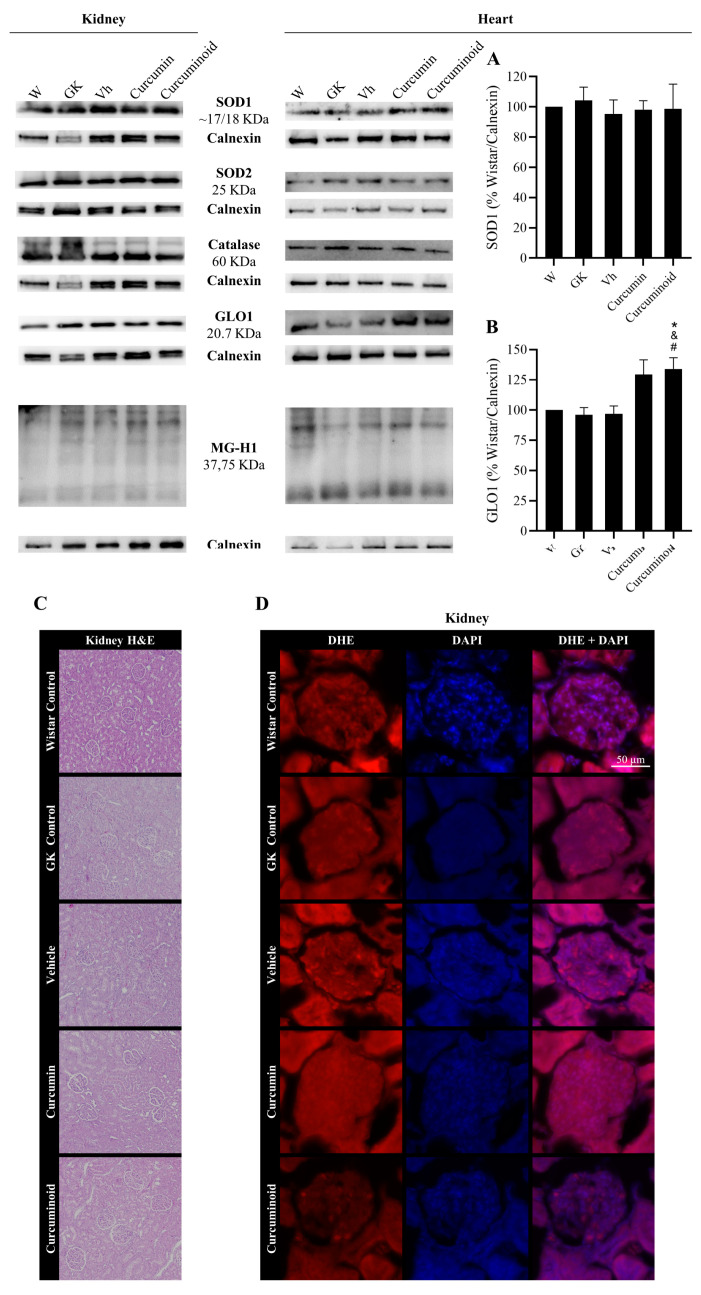
Improved content of the antioxidant enzyme GLO1 (**B**) in the heart after treatment with the curcuminoid, calculated as percentage of Wistar control/Calnexin, assessed by WB; representative WB is shown. SOD1 (**A**) levels were also evaluated in heart by WB. Histological analysis shows hematoxylin–eosin (H&E) staining in kidney (**C**). Oxidative stress was evaluated in the kidney though the determination of the superoxide generation by the DHE dye. Representative kidney sections showing DHE staining (red) (**D**). W: Wistar control; GK: Goto-Kakizaki control; Vh: Goto-Kakizaki submitted to vehicle (DMSO) administration; Curcumin: Goto-Kakizaki submitted to curcumin administration (40 mg/Kg/day, s.c.) and Curcuminoid: Goto-Kakizaki submitted to curcuminoid (52.4 mg/Kg/day, s.c.) administration. Results are expressed as mean ± S.E.M.; vertical bars represent S.E.M.; *n* = 5–9/group; statistical differences were evaluated by Tukey’s test. * vs. W; & vs. GK; # vs. Vh. 1 symbol *p* < 0.05.

**Figure 7 ijms-23-05652-f007:**
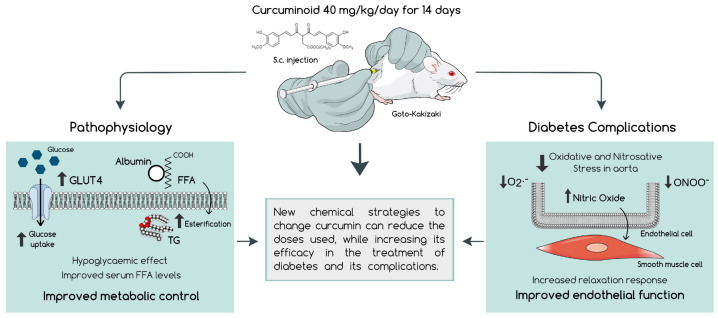
The curcuminoid in a low-dose regimen administered to diabetic rats for 14 days was shown to have a markedly hypoglycaemic effect, accompanied by an improvement of the lipidic profile, which promoted better metabolic control. Furthermore, this new molecule was able to enhance the relaxation response in the aorta, along with a reduction in the vascular oxidative and nitrosative stress, suggesting an improved endothelial function and vascular homeostasis, which may contribute to the prevention of diabetic complications.

**Table 1 ijms-23-05652-t001:** Maximum contractile response (E_max_) ^1^ to acetylcholine of aorta isolated from the different experimental groups.

Groups	E_max_ (%)	a/n
W	51.15 ± 4.52	14/5
GK	44.16 ± 3.11	17/9
Vh	46.02 ± 3.48	11/4
Curcumin (2)	62.22 ± 5.09 ^&&#^	12/4
Curcuminoid (3)	56.25 ± 2.46 ^&^	11/4

^1^ E_max_—maximum relaxation in % of NA induced contraction. A—number of aorta rings; and n—number of animals. Results are expressed as mean ± S.E.M.; statistical differences were evaluated by Kruskal–Wallis test. ^&^
*p* < 0.05 vs. GK; ^&&^
*p* < 0.01 vs. GK; ^#^
*p* < 0.05 vs. Vehicle.

## Data Availability

Not applicable.

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
