# Peer review of "Improvement of Glycaemia and Endothelial Function by a New Low-Dose Curcuminoid in an Animal Model of Type 2 Diabetes"

_ijms, 2022, doi:10.3390/ijms23105652_

Round 1

Reviewer 1 Report

  1. Authors should carefully check the bibliography. Some links are formatted with deviations from the rules for authors (for example, 23 and 24).
  2. Some typos need to be corrected. For example, in lines 415 and 430, instead of NMR, RMN appears.

Author Response

REVIEWER 1:

We are grateful to the reviewer for the comments, which we used to improve the manuscript quality. Please find below a point-by-point response to all the comments, questions and suggestions raised.

  1. Authors should carefully check the bibliography. Some links are formatted with deviations from the rules for authors (for example, 23 and 24).

Thank you very much for the comment, this was corrected.

  1. Some typos need to be corrected. For example, in lines 415 and 430, instead of NMR, RMN appears.

Thank you for the careful review. We corrected the typos of the manuscript.

Reviewer 2 Report

In this manuscript authors analysed the anti-inflammatory and antioxidant potential of a new curcuminoid. Moreover, they also evaluated its  hypoglycaemic and protective vascular effects in type 2 diabetic rats. They found that both curcumin and curcuminoid presented a significant anti-inflammatory effect. Moreover, the curcuminoid had a marked hypoglycaemic effect, accompanied by higher GLUT4 levels in adipose tissue. both curcumin and curcuminoid increased NO-dependent vasorelaxation, but only the curcuminoid exacerbated the response to ascorbic acid, reducing vascular oxidative and nitrosative stress. 

The manuscript is very interesting and generally well written. However, some points need to be improved. In particular:

  • Introduction: Although the authors highlighted the main beneficial effects of curcumin in T2DM, it should also be specified that curcumin can also play an important role in ameliorating Gestational Diabetes Mellitus (GDM) as recently reviewed (PMID: 33477354). This point is very important since GDM can evolve in T2DM after pregnancy (PMID: 35472098). 
  • Line 349: Authors correctly state that endothelial dysfunction is also closely associated to the development of diabetic microvascular complications; however, this is only one of the consequence. Authors should underline that endothelial dysfunction may lead also to many other vascular injuries, including increased permeability (PMID: 34153425; PMID: 33123312).
  • 4.1. Chemicals and Antibodies: Antibodies dilutions must be reported
  • Lines 294-308: Authors discuss the beneficial effect of curcumin in high-fat diet (HFD)-fed mice and rats. However, it should also be said that curcumin can inhibit STAT3 activation in adipocytes (see PMID: 31781039), a key inflammatory pathway associated to insulin resistance (see PMID: 29635003), playing a key function in reducing inflammation and then insulin resistance. This is an important effect of curcumin that should be discussed expecially regarding high-fat diet (HFD). This can further support the very interesting data found by the authors.

Author Response

REVIEWER 2:

In this manuscript authors analysed the anti-inflammatory and antioxidant potential of a new curcuminoid. Moreover, they also evaluated its hypoglycaemic and protective vascular effects in type 2 diabetic rats. They found that both curcumin and curcuminoid presented a significant anti-inflammatory effect. Moreover, the curcuminoid had a marked hypoglycaemic effect, accompanied by higher GLUT4 levels in adipose tissue. Both curcumin and curcuminoid increased NO-dependent vasorelaxation, but only the curcuminoid exacerbated the response to ascorbic acid, reducing vascular oxidative and nitrosative stress.

The manuscript is very interesting and generally well written. However, some points need to be improved.

We are grateful to the reviewer for the positive evaluation of our manuscript and for the careful review. Please find below a point-by-point response to all the comments, questions and suggestions raised.

Comments:

Introduction: Although the authors highlighted the main beneficial effects of curcumin in T2DM, it should also be specified that curcumin can also play an important role in ameliorating Gestational Diabetes Mellitus (GDM) as recently reviewed (PMID: 33477354). This point is very important since GDM can evolve in T2DM after pregnancy (PMID: 35472098).

Thank you for the excellent suggestion, this information was included in the introduction.

Line 349: Authors correctly state that endothelial dysfunction is also closely associated to the development of diabetic microvascular complications; however, this is only one of the consequences. Authors should underline that endothelial dysfunction may lead also to many other vascular injuries, including increased permeability (PMID: 34153425; PMID:33123312).

Thank you for the comment, this information was also included in the manuscript.

4.1. Chemicals and Antibodies: Antibodies dilutions must be reported.

Antibodies dilutions were included in the methods.

Lines 294-308: Authors discuss the beneficial effect of curcumin in high-fat diet (HFD)-fed mice and rats. However, it should also be said that curcumin can inhibit STAT3 activation in adipocytes (see PMID: 31781039), a key inflammatory pathway associated to insulin resistance (see PMID: 29635003), playing a key function in reducing inflammation and then insulin resistance. This is an important effect of curcumin that should be discussed especially regarding high-fat diet (HFD). This can further support the very interesting data found by the authors.

We thank the suggestion; this idea was discussed in the manuscript.

Round 2

Reviewer 2 Report

Manuscript has been significantly improved and can be accepted in the present form.